# Youth daily stressors predict their parents' wellbeing
Melissa A. Lippold [1] ✉, Melissa Jenkins[2], Katherine B. Ehrlich [3], Soomi Lee[4] & David M. Almeida[4]

The experiences of family members are intertwined and the stressors of one family member may crossover to affect the wellbeing of others in the family as well. Prior studies have established that the stress experienced by one marital spouse can affect the wellbeing of their spouse and that parent stress can affect their children's wellbeing. This study used daily diary data from 318 parent-youth dyads (Mean age parent = 41.34, adolescent = 13.18) to examine whether youth daily stressors (i.e., interpersonal conflicts and demands), were associated with parent wellbeing and cortisol levels. Parents report more negative affect, more physical symptoms (i.e., headaches/fatigue/stomach problems), and exhibit higher bedtime cortisol levels on days when youth experience stressors. These effects were consistent across different types of youth stressors, including parent, family, and non-family stressors. Youth stress may have important implications for parent wellbeing.

Stress is not only an individual experience, but a family one as well. Daily stressors (e.g., interpersonal conflicts and demands) that are experienced by one family member can crossover to affect the health and wellbeing of other family members[1,2]. Much of the work on the transmission of daily stress in families focuses on the marital relationship or the effects of parent stress on child wellbeing[3–5]. Less is known about how children's stress may also crossover to affect parent wellbeing[6]. In this paper, we built on prior research to test whether youth daily stressors were associated with their parents' wellbeing, including parent negative affect, physical health symptoms, and stress-related physiology (as assessed by cortisol).

Daily diary approaches are ideal to study youth stress and parent wellbeing, as they minimize recall bias and increase ecological validity[7–10]. A daily diary approach also captures proximal processes, such as how youth stressors are linked to parent affect, physical symptoms, and physiological stress responses *that same day*. Physiological stress responses can be assessed by the hormone cortisol, which marks the functioning of the hypothalamic-pituitary-adrenal (HPA) system. The HPA system exhibits a diurnal rhythm such that cortisol levels peak 30 min after waking and decline over the course of the day, reaching their lowest levels at bedtime. High levels of bedtime cortisol indicate difficulty calming down and restoring from stress at the end of the day and have been linked to negative psychological and physical health outcomes[11].

Adolescence is a critical time to study the effects of child stress on their parents' health because both adolescents and their parents experience high stress during this time[12–14]. Adolescents experience daily stress through experiences and relationships outside of the family, such as challenging relationships with peers, bullying and harassment, increased academic demands, and changing social contexts such as school transitions[12,15]. Stress can also occur within the family, as adolescents' increased need for autonomy can lead to daily disagreements with parents and siblings[16,17]. Parents also experience stress during adolescence. Parents frequently hold negative stereotypes about adolescence as period of family difficulty and report feeling anxiety and worry about their adolescents and less confidence in their ability as parents[18,19].

Family systems theory[1] and emotion transmission models[20,21] posit that the experiences, emotions, and behavior of one family member may affect others within the family through spillover and crossover processes[2,4,22]. Spillover is an intra-personal process that occurs when stress in one domain leads to changes in a person's own behavior, emotions, or attitudes in another domain[2,4]. For example, when an adolescent experiences a stressor with a peer, it may spillover and lead to more irritable and hostile behaviors towards their parents at home[23]. The effects of stress can also crossover to affect the wellbeing of other family members. Crossover is an inter-personal process, when the effects of a family member's stressor can affect the emotional and psychological wellbeing of other members of the family, even though they did not experience the stressor themselves. For example, an adolescent's experience of stress due to conflict with a teacher may crossover to affect their parents' health and wellbeing. Spillover and crossover of stress between family members may occur due to changes in interactions or behaviors or due to empathic responses[24], whereby one person in the family may feel such deep empathy towards the other family member experiencing stress, that it affects their own wellbeing[25]. For example, parents may experience deep empathy, anxiety, and worry about their child's conflict with peers, leading to increased negative parent emotions and physiological stress responses.

[1]The University of North Carolina at Chapel Hill, Chapel Hill, NC, USA. [2]The University of Wisconsin at Madison, Madison, WI, USA. [3]The University of Georgia, Athens, GA, USA. [4]The Pennsylvania State University, State College, PA, USA. ✉e-mail: mlippold@unc.edu

Prior studies have supported crossover models within families[1,2]. However, most of these studies have examined the transmission of stress between marital couples or from parents to their children. For example, prior research has found that stressors experienced by one marital partner can predict the emotions, behaviors, and health of the other marital partner[26,27]. When one person in a marital dyad experiences stress at work for example, their spouse is more likely to experience negative emotions[4,28] and physiological stress responses[29]. Studies have also found that parent stressors, such as work-related stressors[3,30] or marital conflict[31–33] can also crossover to predict child wellbeing[5]. Studies, primarily on younger children, have also found evidence of parent coregulation, such that supportive parental responses to child emotions (and lower invalidation of emotions) are linked to lower child negative emotional responses to stress[34], better child emotion regulation, and lower risk for child behavior problems[35].

Much less is known about how daily youth stressors affect their parents' wellbeing, and whether youth stressors can crossover to alter parents' emotions and physiology. Some studies have found that youth characteristics such as aggressive behavior or conduct problems can affect parenting behavior[36,37] such as parental reductions in monitoring[38] or increases in parent hostility[38,39]. Three recent studies have found that when youth experience high cortisol levels, their parents are more likely to show similar heightened cortisol[6,40,41]. Other studies using lab based methods have found that adolescent negative emotions can predict mother negative emotions[42]. These studies suggest that the effects of youth daily stressors may also crossover to affect parent daily wellbeing. However, the specific associations between youth daily stressors and parents' daily wellbeing have not been thoroughly tested.

Understanding the effects of youth stress on parent wellbeing is critical, as the crossover of youth stress to parents may impact both youth and parent long-term health[43]. In terms of youth wellbeing, parents who experience negative emotional and physiological responses to youth stressors may be less warm, calm, consistent, and supportive in their parent-child interactions[2] which may shut down parent-child communication processes[44,45] and lessen parental support around the child's stressors[12]. Such reductions in effective parenting may make it difficult for youth to calm down after experiencing stressors[10,12] and increase risk for youth internalizing and externalizing problems[43,46–48]. Parents who experience negative emotional and physiological responses to youth stressors may also be at increased risk for their own long-term negative health outcomes[43,49]. Experiencing more stressors and strong reactions to stressors are linked to a host of adult health problems, including lower immunity, increased risk for depression and anxiety, cardiovascular disease, and premature mortality[49,50].

In this study, we built on prior work using parent-child dyads to examine the associations between daily youth stressors and parent wellbeing across an 8-day period using daily diary methods. Parent wellbeing was assessed by negative affect, physical symptoms (i.e., headaches/fatigue/stomach problems), and bedtime levels of cortisol. We examined both within and between-person effects. At the within-person level, we tested whether parents experience lower wellbeing (i.e., more negative affect, more physical symptoms, and higher bedtime cortisol) on days youth when experienced stressors (compared to non-stressor days). We also tested between-person effects, such as whether youth with more stressor days across the 8-day period, on average, were associated with lower parent-wellbeing compared to youth with fewer daily stressors. By including both type of effects, we controlled for stable between-person differences when estimating within-person effects, allowing for stronger inferences[8,10]. We also examined the within-person effects of youth stress on next-day parent wellbeing using lagged models. To gain a comprehensive understanding of youth stressors, we examined these associations for four different types of youth stressors: any stressors, stressors with the focal parent (who participated in the study), stressors with family members (including siblings and parents), and stressors outside of the family (e.g., with friends, at school). We differentiated these types of stressors to discern whether adolescent stressors that could potentially be shared with parents (i.e., parent/family) would have different effects than stressors outside of the family context. This distinction may be especially critical as stressors shared with both parents and adolescents may have especially strong linkages to parent health. We hypothesized that (a.) parents will experience lower well-being (i.e., higher negative affect, more physical symptoms, and higher bedtime cortisol) on days when youth experience stressors and (b.) parents of youth who have a higher average number of stressor days will experience lower average levels of well-being (between-person effects).

## Methods

### Participants and procedure

This study used data from the daily-diary component of the Work, Family, Health Study (WFHS), which examined the effects of a workplace intervention on the health of employees and their families[51]. The WFH study included a subsample of 318 parent-youth dyads who participated in an 8-day daily diary study. On 8 consecutive nightly phone calls, parents and youth provided information on their daily stressors, emotions, and experiences and on four daily diary days (Day 2−5), they also provided saliva samples to assess cortisol levels. During the home interview, parents provided informed consent and youth provided assent to participate in the daily diary and cortisol data collection. Saliva samples were collected from parents at five time points: upon awakening, 30 min after waking, at lunch, before dinner, and before going to bed. During the home interviews, interviewers distributed saliva collection kits, which included 5 salivettes for collecting parent cortisol (5 salivettes/day for 4 days) along with a DVD that demonstrated saliva collection procedures. Parents were instructed to roll a cotton swab across their tongue for two minutes and then return the swab to the tube without touching it and were told not to eat, drink, or brush their teeth within 30 min prior to collection. Participants recorded the time of each saliva sample (using an electronic time stamper) and any medications they were taking on a separate data collection sheet. Participants refrigerated saliva samples after collection and, at the end of the saliva collection period, mailed the samples to the laboratory using prepaid overnight delivery. Upon receipt at the laboratory, saliva samples were weighed and frozen at −80 °C until later assay of cortisol in the Biomarker Core Laboratory at The Pennsylvania State University lab using commercially available EIA kits (Salimetrics, LLC, State College, PA). Assays were run on a rolling basis throughout the entire study period. The assay had a lower limit of sensitivity of 0.003 ug/dL, with average inter- and intra assay covariances of less than 7% and 4%, respectively. Outliers were winsorized such that cortisol values below 0.003 ug/dL were designated as off-the-curve low and were set to the lowest level of sensitivity to the assay. Participants were compensated $150 for completion of the daily diary study. The data collection centers' Institutional Review Boards approved the procedures. The secondary dataset used for this analysis was deemed exempt from the Institutional Review Board at UNC given it was deidentified and did not constitute human subjects research. This study was not preregistered.

The Work, Family, Health study included two samples: employees working in information technology at a Fortune 500 company (IT sample) and a second sample of shift workers in a nursing home (NH). Thus, in order to test our models on a diverse sample and increase power, we combined these samples for this study. In the IT sample ($n = 132$), 45% of employees were women and 55% of the children were girls. Among parents, 78% were college graduates and the average annual income was $116,900 (SD = $26,396). Participants were asked to self-identify their race, ethnicity, and biological sex. The majority of youth were Caucasian (74%), 16% Asian, 2% African American, 2% Pacific Islander, 4% more than one race, and 3% marked other. Of these, 9% indicated they were of Hispanic heritage. For the NH subsample ($n = 186$), 96% of the employees were women and 51% of the children were girls. Among parents, 64% were college graduates and the average annual income was $56,660 (SD = $31,759). Among children in the NH sample, 60% were Caucasian, 14% African American, 3% Asian, 1% American Indian, 7% more than one race, and 15% marked other. Of these, 15% indicated they were of Hispanic heritage. We tested for moderation by sample industry in all analysis to ascertain if study findings were equivalent among those in the IT and NH samples. Because we found no differences by

industry, we present findings from the full, combined sample in this manuscript. In our combined sample, we have 318 dyads that includes 233 mothers and 80 fathers (5 were missing biological sex data; $M_{age}$ = 41.34). Youth ranged from 9 to 17 years of age ($M_{age}$ = 13.18), 167 were daughters and 151 were sons.

## Measures

*Youth Daily Stressors* were measured with an adapted version of the Daily Inventory of Stressful Events[52]. During each call, youth were asked if they had experienced five stressors (0 = *no*; 1 = *yes*) over the past day including three questions where they identified who the stressor was with. An example item is "Did you have any arguments or disagreements with anyone else, other than (your) [TARGET PARENT], since this time yesterday, including other people in your family or people outside your family? (If yes), who did you argue or disagree with?" Three dichotomous scales were created to assess whether youth had experienced stress (0=non stressor day; 1= stressor day) in four categories: (1) any stressors, (2) stressors with the focal parent (who participated in the study), family stressors (e.g., conflicts with any parent and/or siblings) and (3) stressors outside of the family (e.g., conflicts with friends, work/school demands).

*Parent Negative Affect* was assessed using a 10-item subscale adapted from the Positive and Negative Affect Schedule (e.g., "How much of the time today did you feel upset?")[53]. Items were rated on a five-point scale (1 = none of the time to 5 = all of the time)and averaged within each day for use in our analysis. Reliability was calculated at the between- and within-person levels as outlined by Cranford[54]. The between-person reliability was 0.95 and the within-person reliability was 0.73.

*Parent Physical Health Symptoms* was assessed using a ten-item measure adapted from Larsen and Kasimatis.[55] For each item (e.g., headache, very tired, allergies, stomachache, other physical problems or discomforts [not diseases]) parents reported whether they had or had not experienced the symptom that day (0 = no 1 = yes). Responses were summed such that higher scores indicated more physical symptoms that day.

## Parent cortisol measures

Bedtime levels of cortisol (logged) were assessed as levels of cortisol at the end of the day. Cortisol values were converted to nmol/l and natural log transformed before analysis[56]. Given cortisol's diurnal rhythm and prior studies[11], low levels of cortisol at bedtime are considered healthful[57].

*Control Variables* included parent age, youth biological sex (0 = male, 1 = female), parent biological sex (0 = male, 1 = female), parent education (0 = Not a college graduate, 1= College graduate), parent race/ethnicity (0 = White; 1= Non-white), parent body mass index (BMI), type of day (0 = weekday; 1= weekend), sample industry (0 = IT Sample, 1 = NH Sample) and average parent stressors reported that week (using the Daily Inventory of Stressful Experiences)[52]. Cortisol models also controlled for the time of cortisol sample collection, whether or not parents were taking any medications that can influence cortisol (0 = no medications, 1 = 1 or more medications), parent tobacco use (1 = never use; 2=use some days; 3=use most days), and a variable that indicated potential cortisol time non-compliance on a particular day. Parents were given a score of 1 on this noncompliance variable if any of the following conditions were noted during the day of saliva collection: the time difference between samples at wake and 30 min post wake was less than 15 min or greater than 60 min, the parent woke up later than noon, and/or the parent was awake for less than 12 h or more than 20 h. If no compliance issues were noted, the parent received a score of 0 on this compliance variable.

## Statistical analysis

We used Proc Mixed in SAS 9.3 to estimate 2 level multilevel models[58], with days (level 1, within-person) nested within individuals (level 2, between-person). Level 1 included daily measures of youth stressors, which were person-mean centered, and Level 2 included the proportion of youth stressor days across the 8-day study week (grand mean centered) and time-invariant controls (e.g., race, age, biological sex). Models included random intercepts.

$$\text{Level 1: } Parent\ Wellbeing_{ti} = B_{0i} + [B_{1i}(YouthStressorDay_{ti})] + [e_{ti}] \tag{1a}$$

$$\text{Level 2: } B_{0i} = \pi_{00} + [\pi_{01}(AverageStressors_i)] + [u_{0i}] \tag{1b}$$

$$B_{1i} = \pi_{10} + [u_{1i}] \tag{1c}$$

At Level 1 (daily within-person level, Eq. 1(a)), parents *i*'s wellbeing outcomes on day t were modeled as a function of their daily intercept ($B_{0i}$) and daily slope ($B_{1i}$), and residual variance ($e_{ti}$). The daily slope reflects changes in parent wellbeing on days when youth experience a stressor compared to non-stressor days (within-person). At Level 2 (between person-level), the level 1 intercept (Eq. 1b) was modeled as a function of the sample average intercept ($\pi_{00}$), and slope ($\pi_{01}$), as well as random effects ($u_{0i}$). The Level 2 slope ($\pi_{01}$) reflects differences in parent outcomes associated with the cross-time averages of youth stressor days (between-person), that is, differences in parent outcomes, on average, as a function of the proportion of days with youth stressors (as compared to other parents). The level 1 slope (Eq. 1c) was modeled as the sample average daily within-person effect ($\pi_{10}$) and random effects ($u_{1i}$). Lagged models also included prior-day measures of adolescent stress and parent outcomes.

We estimated three models for each outcome variable. First, we tested the main effects of youth daily stressors on parent wellbeing. We tested separate models for 4 types of stressors: any stressors, stressors with the focal parent (who participated in the study), stressors with any family member (parents and siblings), and stressors outside of the family (e.g., with friends/peers). For significant models, we also calculated the percentage change in the outcome between a stressor and non-stressor day to compare effect sizes. In order to address normality, as recommended by other cortisol researchers[11], we natural log transformed our cortisol model before analysis. All three of our outcome variables met criteria for normality as outlined by Kline[59] with a skewness value of <=3 and kurtosis <=10 (skewness: bedtime cortisol= 1.82, negative affect = 2.07, physical symptoms = 1.28; kurtosis: bedtime cortisol = 5.05, negative affect = 5.72, physical symptoms = 1.39). Because our measure of physical symptoms is a count variable, we also conducted sensitivity tests using Poisson models. Results of our physical symptom models were the same for the general linear and Poisson models, so for consistency with other models, we present the results with general linear models (i.e., proc mixed) for all outcomes. Given multiple comparisons, we used the Benjamini Hochberg False Discovery Rate (FDR) procedure to adjust significant tests for multiple comparisons using an FDR rate of 0.05[60] for main effects and interaction terms. For transparency, we report traditional *p* values in all tables, with tests which did not meet FDR-corrected significant levels marked with an asterisk.

## Reporting summary

Further information on research design is available in the Nature Portfolio Reporting Summary linked to this article.

## Results

### Descriptive analysis

Table 1 reports means and within- and between-person correlations for study variables. In terms of between-person correlations, stressors with the focal parent ($r = 0.13$, $p = .02$) and any family members ($r = 0.15$, $p < 0.001$) were positively correlated with parent physical symptoms, and all types of stressors were positively correlated with parent negative affect (any stress $r = 0.26$, $p < 0.001$; parent stress $r = 0.22$, $p < 0.001$, non-family stress $r = 0.13$, $p = 0.02$). In terms of within-person correlations, all types of stressors were significantly correlated with parent physical health symptoms and negative affect (physical symptoms: any stress $r = 0.12$, $p < 0.001$, parent stress $r = 0.07$, $p < 0.001$, family stress, $r = 0.10$, $p < 0.001$, non-family stress $r = 0.07$, $p = 0.001$; negative affect: any stress $r = 0.20$, $p < 0.001$, parent stress

**Table 1 | Means, standard deviations, and correlations between study variables**

| | | *M* | BP SD | WP SD | ICC | 1 | 2 | 3 | 4 | 5 | 6 | 7 | 8 | 9 |
|---|---|---|---|---|---|---|---|---|---|---|---|---|---|---|
| 1 | Any stress | 0.28 | 0.45 | 0.45 | 0.17 | 1.00 | 0.56*** | 0.71*** | 0.50*** | 0.12*** | 0.03 | 0.20*** | 0.02 | 0.06** |
| 2 | Parent stress | 0.12 | 0.31 | 0.31 | 0.13 | 0.71*** | 1.00 | 0.80*** | 0.12*** | 0.07** | 0.26 | 0.10** | 0.01 | .07** |
| 3 | Family stress | 0.17 | 0.37 | 0.37 | 0.15 | 0.81*** | 0.87*** | 1.00 | 0.08*** | 0.10*** | −0.01 | 0.13*** | −0.00 | 0.06** |
| 4 | Non-family stress | 0.09 | 0.28 | 0.28 | 0.10 | 0.54*** | 0.28*** | 0.19*** | 1.00 | 0.07** | 0.03 | 0.34*** | −0.03 | 0.01 |
| 5 | Physical symptoms | 1.47 | 1.64 | 1.64 | 0.51 | 0.11 | 0.13* | 0.15** | −0.01 | 1.00 | 0.03 | 0.33*** | −0.03 | 0.06** |
| 6 | Bedtime cortisol | 1.13 | 0.58 | 0.58 | 0.45 | 0.05 | 0.03 | 0.02 | 0.04 | 0.05 | 1.00 | 0.03 | 0.05 | 0.01 |
| 7 | Negative affect | 1.34 | 0.42 | 0.42 | 0.43 | 0.26*** | 0.22*** | 0.25*** | 0.13* | 0.37*** | 0.03 | 1.00 | −0.02 | 0.06** |
| 8 | Child sex | 52.4%F | – | – | – | −0.01 | −0.03 | −0.03 | 0.06 | −0.05 | 0.05 | −0.03 | 1.00 | −0.02 |
| 9 | Parent sex | 74.4%F | – | – | – | 0.11 | 0.14* | 0.12* | 0.01 | 0.08 | 0.00 | 0.11 | −0.02 | 1.00 |

Within-person correlations are above the diagonal. Between-person correlations are below the diagonal.

*N* = 318, *BP* between-person, *WP* within-person, *SD*s tandard deviation, *ICC* intraclass correlation coefficient, Gender is listed as % female.

*p < 0.05. **p < 0.01. ***p < 0.001.

$r = 0.10$, $p < 0.001$, family stress, $r = 0.13$, $p < 0.001$, non-family stress $r = 0.34$, $p < 0.001$). Most adolescents (78%) reported experiencing stressors at least one day in the past week, with 34% reporting they experienced stressors on three or more days.

## Main effects

Within-person analyses were statistically significant for almost all parental wellbeing outcomes (See Table 2). In terms of effect size for within-person analysis, the percent change in the outcome variable between a stressor and non-stressor day for significant effects ranged from 5 to 40%. That is, parents had higher bedtime cortisol levels ($F(737) = 7.65$, $B = 0.13$, SE = 0.05, $p < 0.001$, ES = 13% change, CI = 0.04−0.22), negative affect ($F(1828) = 56.74$, $B = 0.14$, SE = 0.02, $p < 0.001$, ES = 11% change, CI = 0.10−0.17), and physical health symptoms ($F(1828) = 41.10$, $B = 0.43$, SE = 0.07, $p < 0.001$, ES = 31% change, CI = 0.29−0.55) on days when their children experienced any stressor compared to days with no stressors. These results were consistent across all types of youth stressors and remained significant after controlling for the false discovery rate (FDR), including stressors with the focal parent (negative affect: $F(1827) = 5.27$, $B = 0.06$, SE = 0.03, $p = 0.02$, ES = 5% change, CI = 0.008−0.11; bedtime cortisol: $F(746) = 5.52$, $B = 0.16$, SE = 0.07, $p = 0.02$, ES = 16% change, CI = 0.03−0.29; physical symptoms: $F(1827) = 8.29$, $B = 0.27$, SE = 0.09, $p = 0.004$, ES = 18% change, CI = 0.09−0.45), stressors with family members (negative affect: $F(1826) = 13.86$, $B = 0.08$, SE = 0.02, ES = 6% change, $p < 0.001$, CI = 0.04−0.13; cortisol: $F(732) = 6.48$, $B = 0.14$, SE = 0.05, $p = 0.01$, ES = 14% change, CI = 0.03−0.25; physical symptoms: $F(1826) = 15.76$, $B = 0.32$, SE = 0.08, $p < 0.001$, ES = 22% change, CI = 0.16−0.47), and stressors outside of the family (negative affect: $F(1827) = 58.59$, $B = 0.21$, SE = 0.03, $p < 0.001$, ES = 16% change, CI = 0.15−0.16; physical symptoms $F(1828)$, $B = 0.58$, SE = 0.10, $p < 0.001$, ES = 40% change, CI = 0.38−0.78). However, it should be noted that one model, the effects of non-family stressors on bedtime cortisol, became non-significant once the models were adjusted for the FDR (bedtime cortisol: $F(758) = 4.04$, $B = 0.15$, SE = 0.08, ES = 15% change, $p = 0.04$, CI = 0.004−0.30; adjusted for FDR $p = 0.08$).

There were no statistically significant effects at the between-person level (for negative affect any stressor: $F(334) = 1.39$, $B = 0.08$, SE = 0.07, ES = 6% change, $p = 0.24$, CI = −0.05−0.21; parent stressor $F(340) = 3.11$, $B = 0.18$, SE = 0.10, ES = 14% change, $p = 0.08$, CI = −0.02−0.38; family stressor $F(330)$, = 1.23, $B = 0.10$, SE = 0.09, ES = 7% change, $p = 0.26$, CI = −0.07−0.27; non-family stressor, $F(325) = 2.99$, $B = 0.20$, SE = 0.12, $p = 0.08$, ES = 15% change, CI = −0.03−0.44, for bedtime cortisol: any stressor $F(284) = 0.38$, $B = 0.08$ SE = 0.13, ES = 8% change, $p = 0.54$, CI = −0.17−0.34; parent stressor $F(269) = 0.03$; $B = 0.03$, SE = 0.20, ES = −3% change, $p = 0.87$, CI = −0.36−0.43; family stressor $F(273) = 0.04$, $B = −0.03$, SE = 0.16, $p = 0.84$, ES = 3% change, CI = −0.35−0.29; non-family stressor $F(272) = 3.85$, $B = 0.46$, SE = 0.23, $p = 0.05$, ES = 46% change, CI = −0.001−0.91; Physical symptoms: any stressor $F(315) = 1.44$, $B = −0.34$, SE = 0.28, $p = 0.21$, ES = −111% change, CI = −0.89−0.20; parent stressor $F(309) = 0.01$, $B = 0.05$, SE = 0.42, $p = 0.91$, ES = 26% change, CI = −0.81−0.90; family stressor $F(306) = 0.02$, $B = −0.05$ SE = 0.35, $p = 0.89$, ES = −23% change, CI = −0.75−0.65; non-family stressor: $F(307) = 1.21$, $B = −0.53$, SE = 0.49, $p = 0.27$, ES = −136% change, CI = −1.49−0.42).

## Lagged Effects

Lagged models tested the effects of adolescent stressors on next-day parent health outcomes (controlling for parent same-day outcomes). As shown in Table 3, the within-person effects of adolescent stressors were significantly associated with higher parent negative affect the following day (any stressors: $F(1474) = 20.82$, $B = 0.08$, SE = 0.02, ES = 7% change, $p < 0.001$, CI = 0.05−0.12; parent stressor: $F(1466) = 6.56$, $B = 0.07$, SE = 0.03, $p = 0.01$, ES = 5% change, CI = 0.06−0.12; family stressor: $F(1472) = 8.34$, $B = 0.06$, SE = 0.02, $p = 0.004$, ES = 5% change, CI = 0.02−0.11; non-family stressor: $F(1473) = 15.38$, $B = 0.11$, SE = 0.03, $p < 0.001$, ES = 9% change, CI = 0.06−0.17). Any stressor, parent stressors, and family stressors also had significant effects on next-day parent physical health symptoms (any stressor: $F(1472) = 4.64$, $B = 0.15$, SE = 0.07, $p = 0.03$, ES = 13% change CI = 0.01−0.28; parent stressor: $F(1469) = 4.65$, $B = 0.20$, SE = 0.09. $p = 0.03$, ES = 18% change, CI = 0.02−0.39; family stressor: $F(1468) = 10.90$, $B = 0.26$, SE = 0.08, $p = 0.001$, ES = 23% change CI = 0.11−0.42). However, there were no statistically significant effects of stressors outside of the family on parent next-day physical health symptoms ($F(1469) = 2.57$, $B = 0.17$, SE = 0.10, $p = 0.11$, ES = 14% change, CI = −0.04−0.37) There were also no statistically significant effects of adolescent stress on next-day parent cortisol (any stressor $F(510) = 1.94$, $B = −0.08$, SE = 0.05, $p = 0.16$, ES = −8% change, CI = −0.18−0.03; parent stressor: $F(547) = 0.07$, B = 0.02, SE = 0.08, $p = 0.79$, ES = 2% change, CI = −0.13−0.18; family stressor $F(534) = 0.00$, B = −0.004, SE = 0.07, $p = 0.95$, ES = 0% change, CI = −0.14−0.13; non-family stressor: $F(514) = 1.30$, B = −0.10, SE = 0.09, $p = 0.26$, ES = −10% change CI = −0.27−0.07).

## Discussion

Family systems models posit that the stressors experienced by one family member may affect others in the family as well[1] through crossover processes[2]. Yet, prior literature has been limited and has primarily focused on how stressors crossover between marital partners[31–33] and how parent stressors affect child wellbeing[3,27,30]. This study examined whether the effects of children's stressors crossover to affect parent wellbeing.

We found on days when youth experienced a stressor, their parents experienced lower wellbeing, including greater negative affect, more physical symptoms, such as headaches, fatigue and stomach problems, and

**Table 2 | Same-day effects of youth stressors on parent wellbeing: main effects**

| | Negative affect | | | Bedtime cortisol | | | Physical symptoms | | |
|---|---|---|---|---|---|---|---|---|---|
| | *B* | *SE* | *p* | *B* | *SE* | *p* | *B* | *SE* | *p* |
| **Any stressors** | | | | | | | | | |
| *Fixed effects* | | | | | | | | | |
| Between-person | 0.08 | 0.07 | 0.24 | 0.08 | 0.13 | 0.54 | −0.34 | 0.28 | 0.21 |
| Within-person | **0.14** | **0.02** | **<0.001** | **0.13** | **0.05** | **<0.001** | **0.43** | **0.07** | **<0.001** |
| *Random effects* | | | | | | | | | |
| Intercept | 0.05 | 0.01 | <0.001 | 0.11 | 0.02 | <0.01 | .93 | 0.10 | <0.01 |
| Residual | 0.10 | 0.00 | <0.001 | 0.22 | 0.01 | <0.01 | 1.31 | 0.04 | <0.01 |
| **Parent stressors** | | | | | | | | | |
| *Fixed effects* | | | | | | | | | |
| Between-person | 0.18 | 0.10 | 0.08 | 0.03 | 0.20 | 0.87 | 0.05 | 0.42 | 0.91 |
| Within-person | **0.06** | **0.03** | **0.02** | **0.16** | **0.07** | **0.02** | **0.27** | **0.09** | **.004** |
| *Random effects* | | | | | | | | | |
| Intercept | 0.05 | 0.01 | <0.001 | 0.11 | 0.02 | <0.001 | .92 | 0.10 | <0.001 |
| Residual | 0.10 | 0.00 | <0.001 | 0.22 | 0.01 | <0.001 | 1.33 | 0.04 | <0.001 |
| **Family stressors** | | | | | | | | | |
| *Fixed effects* | | | | | | | | | |
| Between-person | 0.10 | 0.09 | 0.26 | −0.03 | 0.16 | 0.84 | −0.05 | 0.35 | 0.89 |
| Within-person | **0.08** | **0.02** | **<0.001** | **0.14** | **0.05** | **0.01** | **0.32** | **0.08** | **<0.001** |
| *Random effects* | | | | | | | | | |
| Intercept | 0.05 | 0.01 | <0.001 | 0.11 | 0.02 | <0.001 | .92 | 0.10 | <0.001 |
| Residual | 0.10 | 0.00 | <0.001 | 0.22 | 0.01 | <0.001 | 1.33 | 0.04 | <0.001 |
| **Non-family stressors** | | | | | | | | | |
| *Fixed effects* | | | | | | | | | |
| Between-person | 0.20 | 0.12 | 0.08 | 0.46 | 0.23 | 0.05 | −0.53 | 0.49 | 0.27 |
| Within-person | **0.21** | **0.03** | **<0.001** | 0.15 | 0.08 | 0.04* | **0.58** | **0.10** | **<0.001** |
| *Random effects* | | | | | | | | | |
| Intercept | 0.05 | 0.01 | <0.001 | 0.11 | 0.02 | <0.001 | 0.93 | 0.10 | <0.001 |
| Residual | 0.10 | 0.00 | <0.001 | 0.22 | 0.01 | <0.001 | 1.32 | 0.04 | <0.001 |

Significant results are bolded. Separate models were run for each type of stressor and outcome. All models control for parent and child gender, parent age and race, parent education, parent industry, day of week (weekend or weekday), BMI, and average number of parent stressors. Physical symptom models also control for medication use. Cortisol models also control for medication use, tobacco use, time of sample, and time compliance. Coefficients are unstandardized. Sample sizes for each type of stressor are: negative affect: $n = 287$; bedtime cortisol: $n = 277$; physical symptoms $n = 284$. Asterisks indicate coefficients that became nonsignificant when adjusted for the false discovery rate (FDR).

higher (less healthful) bedtime cortisol levels compared to days when youth had no stressors. These same-day findings were consistent across all types of youth stressors, suggesting the crossover of stress from youth to their parents on the same day occurs whether the stressor is with the parent, other family members, or from outside of the family. Notably, these within-person effects of youth stressors on same-day parent wellbeing were found after controlling for stable between-person differences, including other omitted third variables, suggesting that youth stressors are linked to parent wellbeing even when accounting for a host of stable between-person characteristics[8–10].

Adolescent stressors also predicted parent negative affect and physical symptoms the following day. These findings highlight how the effects of child stressors on parents may linger, enhancing their impact and prolonging recovery time for their parents. Given our focus on daily associations, these findings also suggest that many of the effects of youth stressors on their parents' wellbeing are proximal and occur on the same day or the following day.

These findings are in line with recent studies that have found that youth physiological stress responses (i.e., their cortisol levels) predict parent physiology[6,40], as well as studies that have found youth characteristics can predict parenting behavior[36,37]. These findings are also similar to other family crossover processes, such as between marital partners[3,28] and from parents to their children[5,31]. Crossover models that have been tested

primarily on marital couples or from parents to their children should be expanded to consider and test child effects on parent wellbeing. Studies on how child stress affects parents are critical given these crossover effects may affect both youth wellbeing (by reducing effective parenting behavior) and parent wellbeing (as negative affect and physiological stress responses may increase risk for long-term negative health outcomes).

## Limitations
Our findings should be interpreted in light of the study limitations. Although our study provides evidence that youth stressors can crossover to affect parent wellbeing, it is unable to shed light on underlying processes. It is unclear if these crossover processes occur due to changes in youth behavior, parental disengagement, parent empathic stress responses or other cognitive or behavioral factors, and more studies are needed to unpack mediational processes. Although this study reveals short-term negative effects of youth stressors on their parents, it is unclear what the long-term effects of these crossover processes are on youth and parent wellbeing. On the one hand, increased parent negative affect and physiological responses to youth stressors may increase risk for long-term mental and physical health problems in parents. If parental reactions to youth stressors lead to ineffective parenting behaviors, it may also have negative implications for their children. On the other hand, the effects of youth stressors on their parents may

**Table 3 | Next-day lagged effects of youth stressors on parent wellbeing: main effects**

| | Negative Affect | | | Bedtime Cortisol | | | Physical Symptoms | | |
|---|---|---|---|---|---|---|---|---|---|
| | *B* | *SE* | *p* | *B* | *SE* | *p* | *B* | *SE* | *p* |
| **Any stressors** | | | | | | | | | |
| *Fixed effects* | | | | | | | | | |
| Between-person | 0.04 | 0.07 | 0.56 | 0.01 | 0.16 | 0.96 | −0.13 | 0.30 | 0.66 |
| Within-person | 0.09 | 0.02 | <0.0001 | 0.13 | 0.06 | 0.02 | 0.11 | 0.07 | 0.15 |
| Within-person Lagged | **0.08** | **0.02** | **<0.0001** | −0.08 | 0.05 | 0.16 | **0.15** | **0.07** | **0.03** |
| *Random effects* | | | | | | | | | |
| Intercept | 0.05 | 0.01 | <0.0001 | 0.11 | 0.02 | <0.0001 | 0.94 | 0.10 | <0.0001 |
| Residual | 0.08 | 0.00 | <0.0001 | 0.21 | 0.01 | <0.0001 | 1.12 | 0.04 | <0.0001 |
| **Parent stressors** | | | | | | | | | |
| *Fixed effects* | | | | | | | | | |
| Between-person | −0.15 | 0.15 | 0.33 | −0.25 | 0.24 | 0.29 | 0.19 | 0.60 | 0.75 |
| Within-person | −0.00 | 0.03 | 0.90 | 0.15 | 0.09 | 0.08 | −0.03 | 0.11 | 0.80 |
| Within-person Lagged | **0.07** | **0.03** | **0.01** | 0.02 | 0.08 | 0.79 | **0.20** | **0.09** | **0.03** |
| *Random effects* | | | | | | | | | |
| Intercept | 0.05 | 0.01 | <0.0001 | 0.11 | 0.02 | <0.0001 | 0.93 | 0.10 | <0.0001 |
| Residual | 0.09 | 0.00 | <0.0001 | 0.21 | 0.02 | <0.0001 | 1.11 | 0.04 | <0.0001 |
| **Family stressors** | | | | | | | | | |
| *Fixed effects* | | | | | | | | | |
| Between-person | 0.09 | 0.09 | 0.32 | −0.29 | 0.20 | 0.14 | 0.05 | 0.37 | 0.88 |
| Within-person | 0.02 | 0.02 | 0.41 | 0.17 | 0.07 | 0.02 | 0.01 | 0.09 | 0.93 |
| Within-person Lagged | **0.06** | **0.02** | **0.004** | −0.00 | 0.07 | 0.95 | **0.26** | **0.08** | **<0.01** |
| *Random effects* | | | | | | | | | |
| Intercept | 0.06 | 0.01 | <0.0001 | 0.11 | 0.02 | <0.0001 | 0.93 | 0.10 | <0.0001 |
| Residual | 0.09 | 0.00 | <0.0001 | 0.21 | 0.01 | <0.0001 | 1.11 | 0.04 | <0.0001 |
| **Non-family stressors** | | | | | | | | | |
| *Fixed effects* | | | | | | | | | |
| Between-person | 0.15 | 0.13 | 0.24 | 0.45 | 0.27 | 0.09 | −0.31 | 0.51 | 0.54 |
| Within-person | 0.19 | 0.04 | <0.0001 | 0.18 | 0.11 | 0.09 | 0.24 | 0.13 | 0.07 |
| Within-person Lagged | **0.11** | **0.03** | **<0.0001** | −0.10 | 0.09 | 0.26 | 0.17 | 0.10 | 0.11 |
| *Random effects* | | | | | | | | | |
| Intercept | 0.05 | 0.00 | <0.0001 | 0.11 | 0.02 | <0.0001 | 0.93 | 0.10 | <0.0001 |
| Residual | 0.08 | 0.00 | <0.0001 | 0.21 | 0.02 | <0.0001 | 1.11 | 0.04 | <0.0001 |

Significant results are bolded. Separate models were run for each type of stressor and outcome. All models control for parent and child gender, parent age and race, parent education, parent industry, day of week (weekend or weekday), BMI, and average number of parent stressors. Physical symptom models also control for medication use. Cortisol models also control for medication use, tobacco use, time of sample, and time compliance. Physical symptom models also control for medication use. Coefficients are unstandardized. Sample sizes for each type of stressor are: negative affect: *n* = 285; bedtime cortisol: *n* = 249; physical symptoms *n* = 280. The table presents original non-adjusted *p* values. Significant *p* values remained significant after adjusting for the false discovery rate (FDR).

indicate parental responsivity and attunement to their children, which can potentially have positive effects for youth[24]. Further, although this study demonstrated that child stress predicts parent same-day and next-day parent health, this study did not have a sufficient number of measurement occasions to examine the pileup effects of parents' reactivity to child stress over time on longer term health. Prior studies have found that the effects of strong reactions to stress (i.e., high reactivity to stress) can pile up over time and portend risk for longer-term health problems[61]. Future studies are needed to understand the unique role of stressor reactivity pileup on parents' and adolescent long-term health outcomes[61–63].

There were also limitations to our study sample and design. Our study used a combined dataset from two unique samples, which had different demographic profiles. The consistency of our findings across samples suggests these effects may apply to a broad range of individuals. However, our study was not a population-based sample, the children were aged 9–17, and replication on other samples with other ages of children are needed. Although controlling for between-person effects increases our ability to rule out third variables, it is possible that there are other youth or parent factors that could have influenced our findings. For example, although our findings are robust across both family and non-family stressors and control for stable between-person differences, we cannot completely rule out possible recursive effects, such as whether parent characteristics may also affect whether or not children experience a stressor. Studies that examine recursive and bidirectional effects are an important future direction and will require datasets with a larger number of days and/or timepoints. Another limitation is that our assessment of parent bedtime cortisol occurred after youth reported their stressors, but parent reports of negative affect and physical symptoms were gathered at the same time during evening phone calls. We did not have the precise time that the stressor occurred in order to consider stressor timing as a covariate. We were also limited in only having four days of cortisol measurement, which may have reduced our power to detect effects. More studies with additional days and/or shorter timescales like moment-to-moment analysis may be needed to fully examine child stress and parent health and clarify the direction of effect.

Nonetheless, this study makes an important contribution to the literature on spillover and crossover processes in families. It confirms that youth stressors can and do crossover to affect parent wellbeing. When youth experience stressors, not only does it affect their own mood and physiology, it also affects the wellbeing of their parents.

## Data availability
A public-use dataset that reflects the results of these measures is freely available through ICPSR (#36158; https://www.icpsr.umich.edu/web/ICPSR/studies/36158). An extensive, restricted-use dataset is available to researchers who complete the data application and whose project has been approved. For more information see https://workfamilyhealthnetwork.org/data[64].

## Code availability
Code is available via Zenodo at https://zenodo.org/records/11187195[65].

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

## Acknowledgements

The original Work Family Health Study was conducted as part of the Work, Family and Health Network (www.WorkFamilyHealthNetwork.org), which is funded by a cooperative agreement through the National Institutes of Health and the Centers for Disease Control and Prevention: Eunice Kennedy Shriver National Institute of Child Health and Human Development (Grant # U01HD051217, U01HD051218, U01HD051256, U01HD051276), National Institute on Aging (Grant # U01AG027669), Office of Behavioral and Science Sciences Research, and National Institute for Occupational Safety and Health (Grant # U01OH008788, U01HD059773). Grants from the William T. Grant Foundation, Alfred P Sloan Foundation, and the Administration for Children and Families have provided additional funding. The contents of this publication are solely the responsibility of the authors and do not necessarily represent the official views of these institutes and offices. The funders had no role in study design, data collection and analysis, decision to publish or preparation of the manuscript. Special acknowledgement goes to Extramural Staff Science Collaborator, Rosalind Berkowitz King, PhD, and Lynne Casper, PhD, for design of the original Workplace, Family, Health and Well-Being Network Initiative. We also wish to express our gratitude to the worksites, employers, and employees who participated in this research and to Kacey Wyman for her assistance with this paper. Full acknowledgements available at http://www.kpchr.org/wfhn.

## Author contributions

Melissa Lippold conceived of the study, conducted data analysis, and was the primary author of the manuscript. Melissa Jenkins assisted with data analysis and interpretation and aided in the writing of this manuscript. Katie Ehrlich assisted with the conceptualization of the study, interpretation of the findings, and manuscript writing. Soomi Lee assisted with the conceptualization of the study, interpretation of findings, and reviewed all manuscripts. David Almeida assisted with the conceptualization of the study, interpretation of findings, and reviewed all manuscripts.

## Competing interests

The authors declare no competing interests.
