## [Peer Review File · Communications Psychology]

13th Sep 23

Dear Dr Lippold,

We greatly apologise for the delay in processing your manuscript due to difficulties finding suitable reviewers. Thank you for your patience during the process.

Your manuscript titled "Youth Daily Stressors Predict Their Parents' Wellbeing" has now been seen by 2 reviewers, and I include their comments at the end of this message. They find your work of interest, but raised some important points. We are interested in the possibility of publishing your study in *Communications Psychology*, but would like to consider your responses to these concerns and assess a revised manuscript before we make a final decision on publication.

We therefore invite you to revise and resubmit your manuscript, along with a point-by-point response to the reviewers. Please highlight all changes in the manuscript text file.

Editorially, we ask for the revised manuscript to provide more details of methods and a clearer rationale for focusing on adolescence only (see Reviewer #1's comments). Further, please conduct a sensitivity analysis to check for possible recursive dynamics among parents and children and also examine the accumulative effect of stressors as suggested by Reviewer #2.

In addition, please control for multiple comparisons.

A number of the reported results do not reach conventional levels of statistical significance, without any evidence that you a priori set your alpha level to a value other than .05. Please remove claims of significance for any results with p values higher than .05. These results must be reported as non-significant.

In terms of presentation, we agree with Reviewer #1's concerns about the self-referential evaluation in the Discussion. The work should be presented matter of fact, avoiding any claims of perceived importance. These qualitative judgements are best left to the reader. Conversely, please do include a clear discussion of limitations under the heading "Limitations" in the "Discussion" where you include a discussion of the caveats raised by the referees. Because your study did not test an intervention, we discourage discussion of interventions or application.

The manuscript currently does not contain information on informed consent. Please ensure that all information that is requested in the Reporting Summary document is also included in the main manuscript.

Lastly, please ensure the manuscript adheres to our section order (Introduction - Methods - Results - Discussion). Please refer to the checklist and template for which you find hyperlinks below to align your manuscript with our reporting and formatting standards, as this will significantly facilitate processing of your work at the next steps.

Please use the following link to submit your revised manuscript, point-by-point response to the referees' comments (which should be in a separate document to any cover letter) and the completed checklist:

[link redacted]

Please do not hesitate to contact me if you have any questions or would like to discuss these revisions further. We look forward to seeing the revised manuscript and thank you for the opportunity to review your work.

Best regards,

Hannah Hao

Hannah Hao, PhD

Editorial Board Member

Communications Psychology

orcid.org/0000-0002-3342-9132

EDITORIAL POLICIES AND FORMATTING

Editorial Policy: Policy requirements (Download the link to your computer as a PDF.)

Furthermore, please align your manuscript with our format requirements, which are summarized on the following checklist:

Communications Psychology formatting checklist

and also in our style and formatting guide Communications Psychology formatting guide .

* **CODE AVAILABILITY:** All Communications Psychology manuscripts must include a section titled "Code Availability" at the end of the methods section. In the event of publication, we require that the custom analysis code supporting your conclusions is made available in a publicly accessible repository; at publication, we ask you to choose a repository that provides a DOI for the code; the link to the repository and the DOI will need to be included in the Code Availability statement. Publication as Supplementary Information will not suffice. We ask you to prepare code at this stage, to avoid delays later on in the process.

* **DATA AVAILABILITY:**

All Communications Psychology manuscripts must include a section titled "Data Availability" at the end of the Methods section or main text (if no Methods). More information on this policy, is available at <http://www.nature.com/authors/policies/data/data-availability-statements-data-citations.pdf>.

At a minimum the Data availability statement must explain how the data can be obtained and whether there are any restrictions on data sharing. Communications Psychology strongly endorses open sharing of data. If you do make your data openly available, please include in the statement:

We recommend submitting the data to discipline-specific, community-recognized repositories, where possible and a list of recommended repositories is provided at <http://www.nature.com/sdata/policies/repositories>.

If a community resource is unavailable, data can be submitted to generalist repositories such as figshare or Dryad Digital Repository. Please provide a unique identifier for the data (for example a DOI or a permanent URL) in the data availability statement, if possible. If the repository does not provide identifiers, we encourage authors to supply the search terms that will return the data. For data that have been obtained from publicly available sources, please provide a URL and the specific data product name in the data availability statement. Data with a DOI should be further cited in the methods reference section.

REVIEWERS' EXPERTISE:

Reviewer #1 stress and well-being, multilevel analysis

Reviewer #2 parent-child interaction, stress and well-being, multilevel analysis

REVIEWERS' COMMENTS:

Reviewer #1 (Remarks to the Author):

The current study provides an interesting empirical example for how youth daily stress is related to parental well-being (emotional, physical, and biological). This is important work that has been understudied to date. The use of the Work, Family, Health Network Study provided a strong publicly available dataset that can be used for future research. However, there were some weaknesses to the present study that dampened my overall enthusiasm for the manuscript. In particular, there was not enough information pertaining to the daily diary component of the study including how many stressors actually occurred across the eight days. Further, given the outcome variables are vary in type (i.e., categorical, continuous), it was unclear why or how the same modeling approach was appropriate for each one. Below I have listed all my concerns by each section it was a part of.

Abstract

1. It would be useful to see parent average age in addition to youth given the parental well-being is an outcome.
2. Is there clinical or other such implications that you can draw on? The author mentions that youth stress has implications for adult outcomes but don't specify what.

Introduction

1. When speaking about what the author did throughout the introduction, they use present tense, rather than past tense. This is work the author did previously, not what they are currently doing.
2. It seems like there needs to be a stronger argument for why the authors focus on adolescence. How might relationships be changing with parents in adolescence that could inform these associations that are being tested? Given youth spend less time with their parents than children, it is unclear how youth stress informs adult outcomes on a daily level.

3. On the top of page four, would the sentence read, “family members are intertwined” as you are referring to multiple family members well-being?
4. It might be useful for to discuss differences between spillover and crossover, as well as potential differences between crossover and transmission in the transmission of stress in families’ section. These are terms that even researchers get confused sometimes!
5. At the top of page 4, what is meant by crossover processes? Is crossover not a process in and of itself?
6. On page 4, it is unclear what you mean by “Subsequently, the effects the stressor may crossover.” Did the author mean, “the effects of stress” or “the effects of the stressor”?
7. How might regulatory processes (e.g., co-regulation) be informing associations seen in the introduction? Transmission and emotional transmission are pertinent, but similar to marital dyads (e.g., blood pressure) it would be good to know if the parent-child literature has information on co-regulation of emotions/health outcomes.
8. Given you focus on both emotional and physiological outcomes, it may also be pertinent to discuss the implications of social roles and socialization that can inform gender research. For example, in your section on gender differences, the intersection of emotion and physiological reactions may be stronger for women/girls because they were socialized to express emotions at a young age.
9. Based on your literature review, it is unclear why you would disaggregate into different types of stressors. Please provide the rationale to help guide the readers to understand why this is important.

Method

1. It is unclear why the author has put the method section below references. Was it meant to be supplemental material? At the minimum some information pertaining to where to find method information should be included within the document.
2. There was no mention of parent age in the method section. Similar to in the abstract, characterizing average parental age may be pertinent information especially as it is a covariate.
3. Why is only the gender of the youth included as a covariate? What other aspects of youth’s lives could be more pertinent to these associations?
4. Given your outcomes ranged in type (e.g., continuous, summed categorical, biomarker), it is concerning that you used the same models for all variables. Please provide justification for using only general linear mixed models rather than other models (e.g., generalized linear mixed model for physical health symptoms).
5. Was cortisol normally distributed? Why would the author choose to use the raw score rather than say a natural log of cortisol?
6. Medication use could be an important covariate for physical health symptomology as well.

7. Does the daily diary protocol include approximate time with which the youth's stressor occurred? While this is all EOD daily diary, it may be an important covariate to include – especially for the cortisol outcomes.
8. It is unfortunate that the authors chose to only examine bedtime cortisol. There is much more that can be done with five timepoints (e.g., overall daily patterning, total cortisol output) that dampens the enthusiasm for the results.
9. What random effects were included? Both intercept and a random slope for youth stress? Just the intercept? Did models converge with the random slope of stress?

Results

1. The correlation table should also include within-persons correlations where applicable. While standard correlations (e.g., in SAS PROC CORR) cannot account for the repeated assessments, it is possible to estimate correlation coefficients for repeated measures data.
2. It would also be helpful to reiterate in the text that the correlations of mention on page 7 are between-persons only.
3. What was the frequency of stressors that occurred in youths' lives? Did stressors occur on every single day?
4. It would be helpful to have more information pertaining to significant associations in the main effect section. For example, did one type of stressor show larger increases in NA/physical symptoms/cortisol compared to others? Perhaps a figure would be of use.
5. A figure for the significant moderation would also be useful to help guide readers.
6. Please use the appropriate beta symbol. Did the authors standardize their beta coefficients or are these unstandardized effects?
7. I would be curious to see if there was an interaction between parents' gender and youths' gender, it seems like the results by youth gender could be partially explained by whether their target parent is a man or woman.
8. Is it possible to include the parent gender moderation effects within a supplemental table? It would be interesting to see the fixed effects.

Discussion

1. I am not convinced that the phrase "breaks new ground" on page 9 is the best inclusion. The term breaking new ground I believe is an idiom which may not be equally understood in different cultures which may make it difficult to understand the meaning of the sentence.

2. Some of the discussion (e.g., main effects) focuses on reiterating much of the introduction and results. There was no clear explanation discussing why it was expected that all types of stressors reported similar associations, why youth stressors may inform physical or emotional well-being. Could the mechanism be parenting behaviors? Or something else entirely?
3. On page 9, it is a bit of a disservice to argue that the current study provides evidence for less healthy cortisol patterning as the authors only used one time point of cortisol.
4. Given the authors had access to daily level measures, they would be able to examine the lagged effects of associations (i.e., previous day youth stress on parent outcomes); as such it's not clear why the findings only suggest that the effects of youth stress only inform same-day associations.
5. What are the practical implications for the findings? What can clinicians and interventionists take from this work? I think more can be said about this outside of the last sentence of the discussion.

Reviewer #2 (Remarks to the Author):

The primary aim of this study is to examine whether youth experiences of daily stressors influences parental negative affect, physical symptoms, and evening cortisol levels. Gender of youth and parent are also examined as moderators of the carryover effects of youth daily stressors on parental well being. The strengths of this study are: 1. Examination of youth experiences of daily stressors on parental well being in light of previous cross over effects that have largely focused on parental stress cross over effects on either children or partners; 2. Multimethod assessments of parental well being; and 3. Analyses of both within and between youth impacts on parents.

The major challenge in this study is potential recursive effects wherein child impacts on parental well being in turn affect children's experiences of stressors. The authors themselves in the introduction discuss the family systems model which explicitly posits such feedback. One finding herein that argues against such a dynamic was results indicating that the focal parent, family member, or extrafamilial source of daily stress did not matter. Nonetheless I think the authors could run some additional, sensitivity analyses to check possible recursive dynamics among parents and children. For instance, include in a model a covariate of prior day parental outcome for each dependent variable. One could also leverage the longitudinal nature of the data by examining parent (T 1) onto child (T2) onto parent (T3).

The absence of between child effects suggests that most of the child impact on parental well being happens concurrently or at least on the same day with little accumulation over time of repeated child stressors. I am unsure whether it would be worth examining this more directly but given prior research

and theory on the greater potency of accumulated stressors relative to concurrent stressors for well being, some comment is warranted about this topic and how the present data suggest otherwise.

We are grateful to the reviewers for their useful comments on our manuscript and we have used them to improve the manuscript. In this revision we have expanded our introduction regarding the importance of the adolescent time period and spillover and crossover processes. We have added additional analyses include lagged models to assess the effects of youth stress on next-day parent outcomes and Poisson models for our physical symptom models. We have also adjusted our p values to account for multiple comparisons. We have expanded our discussion section on the interpretation of our findings and study limitations. Below we outline our responses in bold text to specific comments.

Reviewer #1 (Remarks to the Author):

The current study provides an interesting empirical example for how youth daily stress is related to parental well-being (emotional, physical, and biological). This is important work that has been understudied to date. The use of the Work, Family, Health Network Study provided a strong publicly available dataset that can be used for future research.

Thank you for your support of our work.

However, there were some weaknesses to the present study that dampened my overall enthusiasm for the manuscript. In particular, there was not enough information pertaining to the daily diary component of the study including how many stressors actually occurred across the eight days. Further, given the outcome variables are vary in type (i.e., categorical, continuous), it was unclear why or how the same modeling approach was appropriate for each one. Below I have listed all my concerns by each section it was a part of.

Thank you for your constructive feedback. Below we respond to your specific points.

Abstract

1. It would be useful to see parent average age in addition to youth given the parental well-being is an outcome.

This information has been added.

2. Is there clinical or other such implications that you can draw on? The author mentions that youth stress has implications for adult outcomes but don't specify what.

In response to the editor comments and in line with journal preference, we have removed discussions of intervention and clinical discussions from this paper.

Introduction

1. When speaking about what the author did throughout the introduction, they use present tense, rather than past tense. This is work the author did previously, not what they are currently doing.

The tense in the introduction has been changed to past tense.

2. It seems like there needs to be a stronger argument for why the authors focus on adolescence. How might relationships be changing with parents in adolescence that could inform these associations that are being tested? Given youth spend less time with their parents than children, it is unclear how youth stress informs adult outcomes on a daily level.

We have added additional information as to why we focus on adolescence, and how both parent and child stress increase during this time period.

3. On the top of page four, would the sentence read, “family members are intertwined” as you are referring to multiple family members well-being?

We have edited this sentence for clarity.

4. It might be useful for to discuss differences between spillover and crossover, as well as potential differences between crossover and transmission in the transmission of stress in families’ section. These are terms that even researchers get confused sometimes!

We have added additional information to differentiate spillover from crossover processes. These two processes reflect different ways stress may be transmitted in families.

5. At the top of page 4, what is meant by crossover processes? Is crossover not a process in and of itself?

6. On page 4, it is unclear what you mean by “Subsequently, the effects the stressor may crossover..” Did the author mean, “the effects of stress” or “the effects of the stressor”?

We have clarified that we mean the effects of adolescent stress may crossover to affect their parents’ wellbeing.

7. How might regulatory processes (e.g., co-regulation) be informing associations seen in the introduction? Transmission and emotional transmission are pertinent, but similar to marital dyads (e.g., blood pressure) it would be good to know if the parent-child literature has information on co-regulation of emotions/health outcomes.

We have added more information about co-regulation to the introduction, including evidence that supportive parent responses to child emotions have been linked to lower child negative emotional responses to stress, better child emotion regulation, and lower risk for child behavior problems.

8. Given you focus on both emotional and physiological outcomes, it may also be pertinent to discuss the implications of social roles and socialization that can inform gender research. For example, in your section on gender differences, the intersection of emotion and physiological reactions may be stronger for women/girls because they were socialized to express emotions at a young age.

This information has been added.

9. Based on your literature review, it is unclear why you would disaggregate into different types of stressors. Please provide the rationale to help guide the readers to understand why this is important.

More information has been added on types of stressors.

Method

1. It is unclear why the author has put the method section below references. Was it meant to be supplemental material? At the minimum some information pertaining to where to find method information should be included within the document.

We have moved the methods section to after the introduction.

2. There was no mention of parent age in the method section. Similar to in the abstract, characterizing average parental age may be pertinent information especially as it is a covariate.

We have added parent age to the abstract and method sections.

3. Why is only the gender of the youth included as a covariate? What other aspects of youth's lives could be more pertinent to these associations?

We agree that there could be other youth factors that could be correlated with parent health outcomes. We now discuss this in the limitations section.

4. Given your outcomes ranged in type (e.g., continuous, summed categorical, biomarker), it is concerning that you used the same models for all variables. Please provide justification for using only general linear mixed models rather than other models (e.g., generalized linear mixed model for physical health symptoms).

General linear mixed models are appropriate for continuous outcomes, which include negative affect and cortisol (which was log transformed before analysis). In response to this comment, we conducted sensitivity tests and ran Poisson models for our physical symptom outcomes, as they are count variables. The Poisson models had the same results as general linear models. For consistency in the paper, we include the general linear model results for all outcomes. We now note this in the paper the results were the same with Poisson models.

5. Was cortisol normally distributed? Why would the author choose to use the raw score rather than say a natural log of cortisol?

We used a log transformed variable for bedtime cortisol. This has been clarified in the manuscript.

6. Medication use could be an important covariate for physical health symptomology as well.

We have added medication as a covariate in our physical health symptom models.

7. Does the daily diary protocol include approximate time with which the youth's stressor occurred? While this is all EOD daily diary, it may be an important covariate to include – especially for the cortisol outcomes.

The dataset does not contain information on when the stressor occurred, which we now add as a limitation. However, the daily diary data on stressors was collected during the evening phone calls (median time 7PM), which was prior to bedtime cortisol collection.

8. It is unfortunate that the authors chose to only examine bedtime cortisol. There is much more that can be done with five timepoints (e.g., overall daily patterning, total cortisol output) that dampens the enthusiasm for the results.

We chose to focus on bedtime cortisol for two reasons. First, our data collection on stressors occurred before bedtime cortisol data collection, which allowed us to ensure the proper temporal ordering to test predictive effects. Given the study design, it would not be possible to ensure that adolescent stressors occurred prior to cortisol output using other aspects of cortisol such as Area Under the Curve or daily slope (which include cortisol collected prior to our assessment of adolescent stress).

9. What random effects were included? Both intercept and a random slope for youth stress? Just the intercept? Did models converge with the random slope of stress?

We have clarified that our models include random intercepts only.

Results

1. The correlation table should also include within-persons correlations where applicable. While standard correlations (e.g., in SAS PROC CORR) cannot account for the repeated assessments, it is possible to estimate correlation coefficients for repeated measures data.

We have added within-person correlations to the correlation table.

2. It would also be helpful to reiterate in the text that the correlations of mention on page 7 are between-persons only.

We have specified that we have included between person correlations in the text.

3. What was the frequency of stressors that occurred in youths' lives? Did stressors occur on every single day?

We have added information on the frequency of adolescent stressors.

4. It would be helpful to have more information pertaining to significant associations in the main effect section. For example, did one type of stressor show larger increases in NA/physical symptoms/cortisol compared to others? Perhaps a figure would be of use.

We have included estimates of the percentage change in the outcome between a stressor and non-stressor day for our significant results. Similar to effect sizes, the percent change allows comparison of the strength of effects between variables.

5. A figure for the significant moderation would also be useful to help guide readers.

There were no significant moderation effects once we adjusted for multiple comparisons using the false-discovery rate. Therefore, we have not added a figure to the paper.

6. Please use the appropriate beta symbol. Did the authors standardize their beta coefficients or are these unstandardized effects?

The coefficients are unstandardized effects, which we now clarify in the table. Because our predictor is coded to indicate a stressor or nonstressor day, the coefficients can be interpreted as the change in the outcome between a stressor versus a nonstressor day.

7. I would be curious to see if there was an interaction between parents' gender and youths' gender, it seems like the results by youth gender could be partially explained by whether their target parent is a man or woman.

We did not include three-way interactions (with both parent and youth gender) given the lower number of fathers in our sample and insufficient power. We identify this as a limitation.

8. Is it possible to include the parent gender moderation effects within a supplemental table? It would be interesting to see the fixed effects.

We have added this as a supplemental table for the review process.

Discussion

1. I am not convinced that the phrase “breaks new ground” on page 9 is the best inclusion. The term breaking new ground I believe is an idiom which may not be equally understood in different cultures which may make it difficult to understand the meaning of the sentence.

We have reworded this sentence to remove the phrase “breaks new ground”.

2. Some of the discussion (e.g., main effects) focuses on reiterating much of the introduction and results. There was no clear explanation discussing why it was expected that all types of stressors reported similar associations, why youth stressors may inform physical or emotional wellbeing. Could the mechanism be parenting behaviors? Or something else entirely?

We have added more information to the introduction and discussion about why different types of stressors may have different effects. We have also expanded our discussion of possible mechanisms, and emphasize the importance of future research in this area. We acknowledge in the limitations section that there may be additional variables, not included here, that influence these processes.

3. On page 9, it is a bit of a disservice to argue that the current study provides evidence for less healthy cortisol patterning as the authors only used one time point of cortisol.

We have clarified that our study informs bedtime cortisol specifically. Given the diurnal profile of cortisol, low cortisol levels of bedtime would indicate the body has recovered from stress.

4. Given the authors had access to daily level measures, they would be able to examine the lagged effects of associations (i.e., previous day youth stress on parent outcomes); as such it's not clear why the findings only suggest that the effects of youth stress only inform same-day associations.

In this revision, we have added lagged models and tested whether adolescent stressors predict next day parent health outcomes. As detailed in our results section, we found consistent evidence that all types of adolescent stressors were associated with next-day parent negative affect and all types of stressors except non-family stressors were associated with next-day parent physical symptoms, but not bedtime cortisol.

5. What are the practical implications for the findings? What can clinicians and interventionists take from this work? I think more can be said about this outside of the last sentence of the discussion.

As per the editor's request, because this is not an intervention study, we have removed discussion of practical and intervention implications from this manuscript.

Reviewer #2 (Remarks to the Author):

1. The primary aim of this study is to examine whether youth experiences of daily stressors influences parental negative affect, physical symptoms, and evening cortisol levels. Gender of youth and parent are also examined as moderators of the carryover effects of youth daily stressors on parental well being. The strengths of this study are: 1. Examination of youth experiences of daily stressors on parental well

being in light of previous cross over effects that have largely focused on parental stress cross over effects on either children or partners; 2. Multimethod assessments of parental well being; and 3. Analyses of both within and between youth impacts on parents.

Thank you for noting the strengths of our study.

2. The major challenge in this study is potential recursive effects wherein child impacts on parental well being in turn affect children's experiences of stressors. The authors themselves in the introduction discuss the family systems model which explicitly posits such feedback. One finding herein that argues against such a dynamic was results indicating that the focal parent, family member, or extrafamilial source of daily stress did not matter. Nonetheless I think the authors could run some additional, sensitivity analyses to check possible recursive dynamics among parents and children. For instance, include in a model a covariate of prior day parental outcome for each dependent variable. One could also leverage the longitudinal nature of the data by examining parent (T 1) onto child (T2) onto parent (T3).

We agree that it is possible that experiences with parents affect child stressors. In this version, we have added lagged models, which examine the effect of youth stressors on next day parent outcomes (while controlling for same day parent outcomes) and increase our confidence in child-driven effects. We agree with the reviewer that the similarity of our findings across types of stressors (including those both within and outside of the family) make the possibility of recursive effects less likely. And further, by controlling for between-person effects, we improve our ability to control for additional third variables. The combination of these factors makes for a compelling case for child effects on their parents.

In this revision, we have added lagged models, to examine effects on next-day parent wellbeing (while controlling for same day wellbeing). In the lagged models, youth stressors predicted next-day parent negative affect and physical symptoms (while controlling for same day parent outcomes). Youth stressors did predict next-day parent negative affect and physical symptoms. However, as we discuss, we did not find lagged effects for bedtime cortisol. Youth stress did not predict higher parent cortisol the following day. Given we only have 4 days of cortisol data (and 8 days of other outcomes), it is possible that we had insufficient power for the cortisol analyses.

We agree with the reviewer that examining within person processes over multiple days (such as day 1 predicting day 2 and day 3) can show an interesting pattern related to recursive and bidirectional relationships. However, answering this question requires a developing a different research question and analytic model which is beyond the scope of this paper. We also do not have a sufficient number of days and statistical power to adequately address this question in our dataset at both the within and between-person levels (i.e., we only have four days of cortisol data). In this revision, we have added this as a limitation and a suggestion for future research.

3. The absence of between child effects suggests that most of the child impact on parental well being happens concurrently or at least on the same day with little accumulation over time of repeated child stressors. I am unsure whether it would be worth examining this more directly but given prior research and theory on the greater potency of accumulated stressors relative to concurrent stressors for well-being, some comment is warranted about this topic and how the present data suggest otherwise.

We have added an additional discussion regarding the cumulative effects of stressors. Our between

person effects suggest that the average levels of stressors across the week are not as consistently linked to parent outcomes as the stressors they experience that same day (i.e., within-person effects) or the following day (lagged effects). This finding is consistent with some other daily diary studies that have found that the number of stressors that occur across the week are not the strongest predictors of health outcomes. Instead, it is strong daily reactions to stressors (called reactivity), and the persistent effects of these daily effects that can lead to chronic stress over time. Several studies have found that individuals who have strong reactions to these daily stressors (i.e., high reactivity) are at increased risk for long-term health problems, and that these daily reactions to daily stressors can “pile up” over multiple days to affect health (Smyth et al., 2013, 2018; Schilling & Deihl, 2014).

We agree that examining pile-up is an important next step in this line of work. The first step, as we focus on in this paper, is to establish whether the effects of youth stressors crossover to affect parent health. Future studies are needed that include more days of data collection to answer questions regarding pileup. In this paper we discuss how an important next step in this line of work is to examine the effects of daily stressor reactivity- that is how stronger daily reactions to child stress and their pileup- can portend risk for longer term health problems.

11th Apr 24

Dear Dr Lippold,

Thank you for your patience during the peer-review process. Your manuscript titled "Youth Daily Stressors Predict Their Parents' Wellbeing" has now been seen by 2 reviewers, and I include their comments at the end of this message.

The reviewers are in principle enthusiastic about your work. However, they also mention a number of concerns. We are very interested in the possibility of publishing your manuscript in *Communications Psychology*, but would like to consider your response to these concerns in the form of a revised manuscript before we make a decision on publication.

In detail, we ask you to address the statistics comment by reviewer 1 and all other minor comments by both reviewers. In sum, we invite you to revise your manuscript taking into account all reviewer and editor comments.

EDITORIAL POLICIES AND FORMATTING

You will find a complete list of formatting requirements following this link:

<https://www.nature.com/documents/commsj-style-formatting-checklist-review-perspective.pdf>

Please use the checklist to prepare your manuscript for resubmission.

* **TRANSPARENT PEER REVIEW:** *Communications Psychology* uses a transparent peer review system. This means that we publish the editorial decision letters including Reviewers' comments to the authors and the author rebuttal letters online as a supplementary peer review file. We publish these records for all accepted manuscripts. However, on author request, confidential information and data can be removed from the published reviewer reports and rebuttal letters prior to publication. If your manuscript has been previously reviewed at another journal, those Reviewers' comments would not form part of the published peer review file.

If you have any questions about any of our policies or formatting, please don't hesitate to contact me.

Please use the following link to submit your revised manuscript and a point-by-point response to the referees' comments (which should be in a separate document to any cover letter):

[link redacted]

We hope to receive your revised paper within 12 weeks; please let us know if you aren't able to submit it within this time so that we can discuss how best to proceed. If we don't hear from you, and the revision process takes significantly longer, we may close your file.

We understand that due to the current global situation, the time required for revision may be longer than usual. We would appreciate it if you could keep us informed about an estimated timescale for resubmission, to facilitate our planning. Of course, if you are unable to estimate, we are happy to accommodate necessary extensions nevertheless.

Please do not hesitate to contact me if you have any questions or would like to discuss these revisions further. We look forward to seeing the revised manuscript and thank you for the opportunity to review your work.

Best regards,

Hannah Hao, PhD

Editorial Board Member

Communications Psychology

orcid.org/0000-0002-3342-9132

REVIEWERS' EXPERTISE:

Reviewer #1 stress and well-being (incl saliva cortisol analysis) and multilevel analysis

Reviewer #2 stress and well-being (incl saliva cortisol analysis), parent-child interaction and multilevel analysis

REVIEWERS' COMMENTS:

Reviewer #1 (Remarks to the Author):

The authors addressed all main points in the original review well. The manuscript provides interesting results, and I personally appreciate the inclusion of the lagged effects. It would be interesting to see bi-directional effects (does negative affect of parents inform youth stress, for example?), but this would be too much for an already complex and full manuscript. There are a few more points I would like addressed, but they are minor in nature. See below.

Major Points:

1. In the discussion, the authors state that day when youth experience stressors, there was a lower (healthful) level of bedtime cortisol levels for parents compared to when no stressor occurred. However, in the results, they report a 13% increase in bedtime cortisol for parents when a youth stressor occurred compared to when there was no stressor. This discrepancy is concerning but mistakes are made all the time. That said, please double check manuscript and statistical analyses for any additional potential errors and possible interpretation.

Minor Comments:

1. Avoid using biased language in the section on gender differences. In line with APA guidelines for language regarding gender, the discussion of socialization and cultural roles assigned or developed in the U.S. would equate to the use of the term "Women" rather than "females."
2. In Table 3, there was one number that wasn't aligned with the others.
3. In Table 4, there were asterisks indicating that the effects were significant but became N.S. with the FDR adjusting. Should these have been bolded?

Signed,

Dakota D. Witzel, PhD

Reviewer #2 (Remarks to the Author):

The authors have been responsive to the reviews they received and I recommend publication.

One small point in the revised Discussion section on p. 17 They note that their absence of gender interactions on the child to parent spillover is consistent with mixed findings in the literature. They need a cite(s) for the reference to prior literature.

I missed the first time around that the authors have enough data to look at additional cortisol outcomes such as AUC or the awakening response or the AUC after the morning response. In their reply to the other reviewer on this point, the authors note that because of when daily stressors was recorded, they can only be certain re: the bedtime measure followed any stressors experienced during the day. I think that is a reasonable reply, but wonder if the specific stressors assessed for each youth, would be sufficient to actually determine the timing of stressors for at least some youth? If yes, then it might be worth exploring AUC without the awakening response given that the morning measures clearly precede what happens during the day. I recognize this is speculative and might not even be feasible, but since they have repeated measures of cortisol, the reviewer is correct that in principle more could be examined with the cortisol measure.

Reviewer 1:

Major Points:

1. In the discussion, the authors state that day when youth experience stressors, there was a lower (healthful) level of bedtime cortisol levels for parents compared to when no stressor occurred. However, in the results, they report a 13% increase in bedtime cortisol for parents when a youth stressor occurred compared to when there was no stressor. This discrepancy is concerning but mistakes are made all the time. That said, please double check manuscript and statistical analyses for any additional potential errors and possible interpretation.

We have corrected this typo in the discussion section. The discussion clarifies that on days when youth experience stressors, there was a higher level of bedtime cortisol. We have also double checked the data and manuscript.

Reviewer #1 (minor comments)

Minor Comments:

1. Avoid using biased language in the section on gender differences. In line with APA guidelines for language regarding gender, the discussion of socialization and cultural roles assigned or developed in the U.S. would equate to the use of the term “Women” rather than “females.”

2. In Table 3, there was one number that wasn’t aligned with the others.

3. In Table 4, there were asterisks indicating that the effects were significant but became N.S. with the FDR adjusting. Should these have been bolded?

We have made the necessary changes above. We only bolded significant results that held after the FDR adjustment.

Reviewer #2:

Remarks to the Author:

The authors have been responsive to the reviews they received and I recommend publication.

Thank you for support of our work

One small point in the revised Discussion section on p. 17 They note that their absence of gender interactions on the child to parent spillover is consistent with mixed findings in the literature. They need a cite(s) for the reference to prior literature.

We have removed this sentence from the manuscript.

I missed the first time around that the authors have enough data to look at additional cortisol outcomes such as AUC or the awakening response or the AUC after the morning response. In their reply to the other reviewer on this point, the authors note that

because of when daily stressors was recorded, they can only be certain re: the bedtime measure followed any stressors experienced during the day. I think that is a reasonable reply, but wonder if the specific stressors assessed for each youth, would be sufficient to actually determine the timing of stressors for at least some youth? If yes, then it might be worth exploring AUC without the awakening response given that the morning measures clearly precede what happens during the day. I recognize this is speculative and might not even be feasible, but since they have repeated measures of cortisol, the reviewer is correct that in principle more could be examined with the cortisol measure.

We did not have the data structure necessary to ensure the proper temporal precedence or to ascertain the precise timing of stressors.